# Association between Dietary Acid Load and Chronic Kidney Disease in the Chinese Population: A Comprehensive Analysis of the China Health and Nutrition Survey (2009)

**DOI:** 10.3390/nu16152461

**Published:** 2024-07-29

**Authors:** Shurui Wang, Xiaohong Fan, Xixi Zheng, Peng Xia, Haiou Zou, Zhaofeng Zhang, Limeng Chen

**Affiliations:** 1School of Nursing, Chinese Academy of Medical Sciences and Peking Union Medical College, Beijing 100730, China; shurui20200101@163.com (S.W.); haiou5275@163.com (H.Z.); 2Department of Nephrology, State Key Laboratory of Complex Severe and Rare Diseases, Peking Union Medical College Hospital, Chinese Academy of Medical Science and Peking Union Medical College, Beijing 100730, China; fxhcmu@163.com (X.F.); zxxpumch@163.com (X.Z.); 7-xp@163.com (P.X.); 3Department of Nutrition & Food Hygiene, School of Public Health, Peking University Health Science Center, Haidian District, Beijing 100191, China

**Keywords:** chronic kidney disease, DAL, PRAL, NEAP, CHNS

## Abstract

Background: Dietary acid load (DAL) is closely related to several chronic diseases. However, the link between DAL and chronic kidney disease (CKD) remains scarce and without data from the Chinese populations whose diet is quite different from people in Western countries. Methods: This study evaluated DAL by potential renal acid load (PRAL) and net endogenous acid production (NEAP). We clarified the relationship between DAL and CKD by logistic regression analysis based on data from the China Health and Nutrition Survey (CHNS). Results: The final analysis included 7699 individuals, of whom 811 (11.44%) were CKD patients. Although there was no notable link between PRAL and CKD, higher NEAP levels were independently correlated with CKD. As NEAP values rise, so does CKD prevalence. This trend remains highly significant even after adjustments. In subgroup analyses, the relationship between NEAP and CKD was more consistent in the elderly and subjects with a waistline of less than 82 cm and those without diabetes and heart disease. RCS analysis further confirmed the clear linear relationship between the OR of CKD and NEAP score. Conclusions: This study highlighted that higher NEAP was positively correlated with the risk of CKD.

## 1. Introduction

In the preceding twenty years, the incidence and mortality rates of CKD have gradually increased worldwide. According to *The Lancet*, in 2019, there were 697.5 million patients with CKD globally, constituting 9.1% of the world’s population from 1990 to 2017 [1]. This prevalence surpasses non-communicable chronic diseases such as metabolic disorders, cardio-cerebrovascular disease, bronchopneumonia, and gastroenteritis [2]. Even more remarkably, 90% of adults affected by CKD are unaware of their condition, and half of those with diminished kidney function who are not undergoing dialysis are oblivious to the presence of CKD. Therefore, the early blocking of the progress of CKD is beneficial to reducing the morbidity and mortality of ESRD.

Recently, besides pharmacological therapies, medical nutrition and dietary intervention were recognized as effective strategies for preventing and treating CKD. Many studies have confirmed that diet interventions are convenient, inexpensive, and practical methods in CKD therapies [3,4]. The appropriate intake of proteins, phosphorus, sodium, and potassium could effectively prevent or improve the symptoms of insufficient renal function, diminishing proteinuria, maintaining good nutritional status, and avoiding malnutrition and metabolic disorders [5,6,7]. These metabolic disturbances comprise a cluster of dysfunctions, including protein-energy wasting, malnutrition, and progressive decreases in muscle function, all of which might accelerate CKD progression and heighten the incidence of complications, including frailty, cardiovascular disease, disturbed blood control, infectious complication, and the mortality of CKD and ESRD patients [8]. Intuitively, the disorders of acid–base balance are the most common complication of CKD and aggravate gastrointestinal symptoms [9]. There is compelling evidence that DAL is the primary determinant of the acid–base balance and systemic pH clinical research has shown a positive correlation between DAL and metabolic syndromes, chronic inflammation, hypertension, and hyperuricemia [10,11,12,13], which all contribute to CKD progression.

In animal experiments, DAL is associated with increased protein intake, leading to a reduction in renal medullary and muscle pH accompanied by an elevation in ammonia concentration. Clinical studies indicated that diets rich in fruits and vegetables buffered DAL, attenuating kidney injury and slowing GFR decline [14,15]. Tanushree demonstrated that the prevalence of kidney failure was threefold higher for people with higher DAL than those with lower DAL [16]. Crews suggested that the effect of DAL on CKD patients may vary by ethnicity and race-related perceptions, which suggests that these clinical outcomes in Western countries, as described in previous studies, may not be beneficial to the Asia population [17]. Unfortunately, no investigation has focused on the relationship between DAL and CKD in the Chinese population.

To fill this gap, we conducted this study based on cross-sectional data from the CHNS, which collected extensive community-level data from 1989 to 2015 in China [18]. Therefore, this study aims to determine the association between DAL and CKD in a large Chinese population. The results will provide new perspectives on the diagnosis and prognosis of CKD patients while also developing innovative strategies and public health policies to enhance the quality of life and life expectancy of the elderly population.

## 2. Materials and Methods

### 2.1. Data Collection and Participants

This study utilized openly accessible data from the CHNS. This study utilized a multi-stage random clustering method to gather comprehensive data on the socioeconomic status, healthcare utilization, dietary habits, and nutritional status of a consistent cohort over time. Finally, we selected approximately 7200 households, with a total population exceeding 30,000 individuals.

We gathered essential particulars, health-related data, socioeconomic status, diet and exercise information derived from the 2009 database, and exclusively biological data. A total of 7749 participants were recruited from 18,805 individuals in this study, excluding individuals with incomplete information, missing variables, no EGFR data, and those under 18 years old (Figure 1).

All the participants provided informed consent. The study was approved by the Carolina Population Center at the University of North Carolina and the National Institute of Nutrition and Health Research at the Chinese Center.

### 2.2. Predictors—Dietary Acid Load Estimations

The dietary information was gathered by instructing the participants to record their daily food intake. Strictly speaking, the participants should document the types and quantities of food they consume within 24 h, covering two workdays and one weekday. The nutrient and energy intake were measured based on the Chinese Food Composition Tables as follows [14,19]:PRAL (mEq/d) = 0.4888 × protein (g/d) + 0.0366 × phosphorus (mg/d) − 0.0205 × potassium (mg/d) − 0.0125 × calcium (mg/d) − 0.0263 × magnesium (mg/d); 
NEAP (mEq/d) = (54.5 × protein (g/d) ÷ potassium (mEq/d)) − 10.2.

### 2.3. Outcomes—Definition of CKD

The estimated glomerular filtration rate (eGFR) was determined using the CKD Epidemiology Collaboration formula [20]. CKD was characterized by an eGFR < 60 mL/min/1.73 m^2^ [21].

### 2.4. Statistical Analysis

Continuous variables were summarized as mean (SD) or median (IQR), and categorical variables as frequencies. Group comparisons utilized Student’s *t*-test, Fisher’s exact test, or chi-square test, as appropriate.

Both univariate and multilevel logistic regression analyses were used to identify the association between CKD and DAL. We categorized the individuals into four groups according to the quartile distribution (Q1, Q2, Q3, and Q4); the odds ratios (ORs) were calculated based on the quartiles of the DAL score. Then, we conducted five models; model 1: unadjusted for covariates; model 2: adjusted for demographic variables; model 3: adjusted for hypertension, diabetes, heart disease, and fraction. Model 4: adjusted for hypertension, diabetes, heart disease, and fraction based on model 2; model 5: adjusted for energy, fat, cholesterol, CHO, and water. We also performed a subgroup analysis according to gender, age, and the absence of comorbidities (diabetes, hypertension, and heart disease). In the end, a restricted cubic spline (RCS) analysis was employed to demonstrate the relationship between CKD and DAL. Significance was set at *p* < 0.05.

## 3. Results

### 3.1. Baseline Characteristics

In this study, a total of 7749 individuals were included (Figure 1), and the general characteristics of the participants are presented in Table 1. In all, 895 patients (11.55%) were diagnosed with CKD with a mean age of 50.2 ± 15.0 years, and 22.7% had a high school or higher education. About one-third of them had a history of tobacco smoking (31.06%) or alcohol intake (32.68%). The underlying diseases included hypertension (13.06%), diabetes (2.79%), and cardiovascular disease (0.94%).

The average PRAL and NEAP were 21 (interquartile range, 13–28) mEq/d and 78 (65–92) mEq/d, with higher NEAP in the CKD patients than the non-CKD participants (80 (67, 95) mEq/d vs. 77 (65, 92) mEq/d, *p* = 0.001, Table 1)). The mean energy intake was approximately 1656 (1354, 1985) Kcal/day and less potassium (K), phosphorus (P), magnesium (Mg), and higher levels of fat and calcium (Ca) intake in the CKD patients than in the non-CKD population, but no difference in protein, phosphorus (P), and cholesterol between the two groups (Table 2).

### 3.2. Association between DAL and CKD

There was no significant association between CKD and PRAL in the five multivariate logistic regression models adjusted by different covariates (Table 3). The relationship between NEAP and CKD was shown in five logistic regression models, and an elevated level of NEAP was an independent association factor for CKD risk in the adjusted models 2, 3, 4, and 5 (Table 4).

### 3.3. Subgroup Analysis for the Association between DAL and CKD

In the subgroup analyses, there was no significant association between PRAL and the risk of CKD with the factors (gender, age, waist, BMI, and comorbidities, including diabetes, hypertension, and heart disease) responsible for potential heterogeneity (Appendix A). However, we observed that the positive correlation between NEAP and CKD appeared stronger in those aged ≥ 60 years with lower waist circumference and BMI < 24 kg/m^2^. Further, in the subgroup without heart disease (Q3: OR 1.30, 95% CI 1.01–1.69; Q4: OR 1.32, 95% CI 1.00–1.75) and diabetes 1.32 (95% CI, 1.01–1.72) and 1.34 (95% CI, 1.01–1.78) participants were more likely to suffer from CKD after adjusting for all the confounders, but not in subjects with or without cardiovascular disease (Appendix A).

### 3.4. RCS Analysis of DAL and Risk of CKD

The RCS analysis showed non-linear associations between the OR values of CKD and PRAL (*p* > 0.05; Figure 2A) based on the multivariate logistic proportional hazards model. However, the NEAP score increased the risk of CKD (*p* < 0.05), which revealed a clear linear relationship between the OR values of CKD and NEAP with the cutoff of 77.99 mEq/day (Figure 2B).

### 3.5. Dietary Intakes and CKD Risk in an Energy-Adjusted Model

In the energy-adjusted model, we found that protein and fat increased the risk of CKD by 16% (OR: 1.16, 95% CI: 1.06–1.27) for each 10 g increased intake, and each 100 g fat and carbohydrate intake may increase the risk of CKD by 95% (OR: 1.95, 95% CI: 1.41–2.68) and 21% (OR: 1.23, 95% CI: 1.08–1.4) respectively. In addition, calcium was also observed to increase the risk of CKD by 20% for each 100 g increased intake. However, for every 100 mg reduction in energy-adjusted phosphorus and magnesium intake, the risk of CKD may decrease by 17% and 18%, respectively (Figure 3).

## 4. Discussion

As the first multicenter national cross-sectional study in China, we first identified a significant correlation between elevated levels of NEAP and an increased risk of CKD. In the subgroup analyses, the correlation between NEAP and CKD was more consistent in the elderly, waist < 82 cm, and those without diabetes, and heart diseases. The RCS analysis further indicated a clear linear relationship between the OR of the CKD and NEAP score. In addition, we confirmed that higher consumption of carbohydrates, fat, and protein was correlated with an increased risk of CKD. 

Dietary intake is a crucial factor influencing the acid–base balance in individuals with chronic diseases. A substantial body of research has developed various methods to estimate DAL based on dietary intake measurements including endogenous acid production, gastrointestinal (GI) alkali absorption, NEAP, net acid excretion NAE, and PRAL [22,23,24,25]. The most prevalent methodologies for assessing acid–base balance are NEAP and the diet-dependent component of net acid excretion, referred to as PRAL [25,26], both of which assess acid–base status based on food components such as calcium, phosphorus, magnesium, protein, and potassium. NEAP, calculated from the ratio of dietary protein to potassium content, indicates the equilibrium between acid and base precursors in the body. In this study, we first observed that NEAP was positively significantly correlated with the prevalence of CKD in the Chinese population. It filled the gap of Chinese data in this controversial field. This conclusion was inconsistent with previous studies from European and American countries and some Asian regions. Clinical research from Johns Hopkins University suggested that a higher NEAP was correlated with a higher CKD incidence independent of sociodemographic, clinical, and lifestyle factors in a population-based sample [27]. The African American Study drew similar conclusions that higher estimated NEAP predicts a faster decline in directly measured I125 iothalamate GFR [28]. Kabasawa and colleagues from Japan demonstrated that elevated NEAP levels were correlated with the presence of albuminuria, which serves as a marker for CKD [29]. In Banerjee’s cross-sectional cohort study, higher NEAP value was strongly associated with progression to ESRD (HR 3.04 (1.58 to 5.86)) [16]. However, Pike and colleagues found no associations between the NEAP score and CKD progression in any APOL1 genotype [30]. Scialla and colleagues observed that a higher NEAP was correlated with the increased consumption of meat and poultry and decreased intake of fruits and vegetables, but it was not linked to the progression of CKD. However, elevated NEAP estimated from urine biomarkers (NEAP_Urine_) was correlated with a higher risk of CKD in individuals without diabetes [31]. 

Then, we further conducted sensitivity analyses to see if our results were robust. In this study, we found that older adults and individuals with smaller waist circumferences tended to have higher acid load. Elderly individuals often experience a reduction in muscle mass, insufficient nutritional intake, decreased metabolic efficiency, weakened buffering system function, and chronic diseases [32,33,34]. These factors contribute to a diminished ability to buffer acids, resulting in an increased acid load in the body [34,35]. Additionally, people with smaller waist circumferences often have less muscle mass and fat reserves, which can lead to disturbances in acid–base balance, and are more susceptible to the effects of NEAP [36,37]. These findings underscore the significance of body composition and muscle mass in evaluating dietary acid load and its health repercussions, particularly in vulnerable elderly groups. In addition, we also found that CKD patients without diabetes and heart disease may be more susceptible to the effects of NEAP. This result is consistent with a previous cross-sectional study in Korea, which suggested that the relationship between eNEAP and CKD appeared stronger in those without obesity, diabetes, and heart disease [38]. These studies demonstrated that in CKD patients without diabetes or heart disease complications, effectively controlling DAL levels contributes significantly to reducing the risk of CKD. 

We also confirmed that the people with higher intakes of carbohydrates, fat, and protein might be at an increased risk of CKD, consistent with the majority of the previously published studies [7]. High protein, fat, and carbohydrate load might contribute to proteinuria and the disorder of electrolytic and acid–base balance, resulting in glomerular hyperfiltration and CKD progression [39,40,41,42,43]. Rebholz et al. demonstrated that a DASH-style diet, a dietary pattern high in fruits, vegetables, low-fat, and carbohydrates, was a protective factor for CKD, and this pattern gives evidence for dietary approaches to the prevention and treatment of CKD [44]. Similar findings from research on the Mediterranean diet (MD), which is characterized by the regular intake of plant-based foods, indicate that it can lower the levels of homocysteine, microalbuminuria, total cholesterol, and phosphorus while increasing vitamin B12 levels in the early stages of CKD [45]. In conclusion, a traditional diet rich in plant-based proteins supports glomerular hemodynamics and offers protective benefits for renal function [46].

This study had several limitations. Firstly, as a cross-sectional study, it could not prove the cause-and-effect relations between DAL and CKD. Secondly, although PRAL and NEAP are widely used indicators of overall DAL, it is calculated by diet but not from the directly measured data. Thirdly, since dietary assessment came from the recall of food intake by the participants, recall bias could not be avoided, although we tried several ways to minimize it. Fourthly, due to the presence of some missing data (sampling framework, weight information, serum calcium, and serum phosphorus) in this population-based study, the conclusions may have certain limitations when applied to a broader population, particularly the Chinese population.

## 5. Conclusions

In conclusion, this study is the first to evaluate the relationship between DAL and CKD in the Chinese population. The graphical summary of the study is shown in Figure 4. It concluded that higher NEAP positively correlated with the prevalence of CKD, either with or without covariate adjustment. It may provide novel insights and strategies to improve the standard of living and life expectancy of the elderly population.

## Figures and Tables

**Figure 1 nutrients-16-02461-f001:**
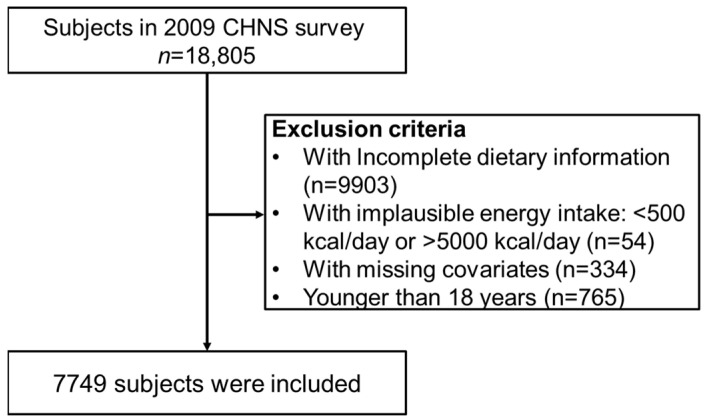
Flow chart of research participant selection.

**Figure 2 nutrients-16-02461-f002:**
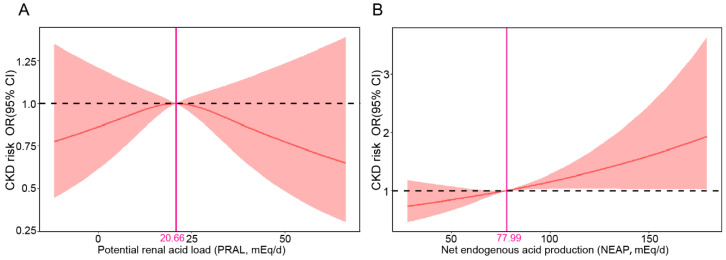
RCS of the correlation between the CKD risk (ORs) and PRAL (**A**) and NEAP (**B**). Pink solid line indicates cut-off value.

**Figure 3 nutrients-16-02461-f003:**
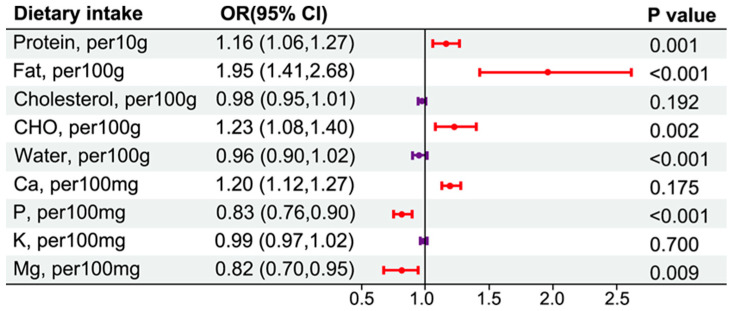
Forest plot of the relationship between dietary intakes and CKD risk (energy-adjusted model).

**Figure 4 nutrients-16-02461-f004:**
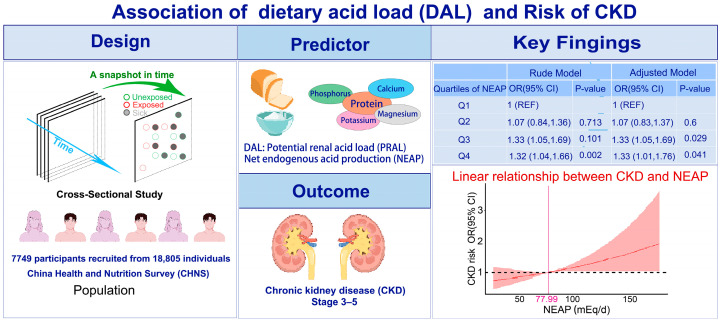
Graphical summary of this study.

**Table 1 nutrients-16-02461-t001:** Subject characteristics based on CKD status.

Variable		Overall (*n* = 7749)	Non-CKD (*n* = 6854)	CKD (*n* = 895)	*p*
Age (y)		50.2 ± 15.0	47.9 ± 13.9	67.9 ± 10.8	<0.001
Gender	Male	3639 (46.96%)	3281 (47.87%)	358 (40%)	<0.001
Education	Primary	3417 (44.1%)	2785 (40.63%)	632 (70.61%)	<0.001
	Middle	2573 (33.2%)	2436 (35.54%)	137 (15.31%)	
	High	886 (11.43%)	834 (12.17%)	52 (5.81%)	
	Vocational	520 (6.71%)	468 (6.83%)	52 (5.81%)	
	University	353 (4.56%)	331 (4.83%)	22 (2.46%)	
Marry	Unmarried	1163 (15.01%)	914 (13.34%)	249 (27.82%)	<0.001
	Married	6586 (84.99%)	5940 (86.66%)	646 (72.18%)	
Location	Rural	4758 (61.04%)	4316 (62.97%)	442 (49.39%)	<0.001
	Urban	2991 (38.6%)	2538 (37.03%)	453 (50.61%)	
Alcohol	Yes	2532 (32.68%)	2359 (34.42%)	173 (19.33%)	<0.001
Smoking	Yes	2407 (31.06%)	2176 (31.75%)	231 (25.81%)	<0.001
Hip (cm)		94.4 ± 7.8	94.4 ± 7.8	94.0 ± 8.5	0.056
Waist (cm)		82.6 ± 10.3	82.4 ± 10.2	83.8 ± 10.7	<0.001
BMI (kg/m^2^)		23.3 ± 3.4	23.3 ± 3.4	23.3 ± 3.8	0.410
BMI (kg/m^2^)	<18.5	490 (6.32%)	411 (6.0%)	79 (8.83%)	0.005
	(18.5, 23.9)	4186 (54.02%)	3717 (54.23%)	469 (52.4%)	
	(24, 27.9)	715 (9.23%)	622 (9.07%)	93 (10.39%)	
	≥28	2358 (30.43%)	2104 (30.7%)	254 (28.38%)	
Hypertension	Yes	1012 (13.06%)	732 (10.68%)	280 (31.28%)	<0.001
Diabetes	Yes	216 (2.79%)	152 (2.22%)	64 (7.15%)	<0.001
Heart Disease	Yes	73 (0.94%)	44 (0.64%)	29 (3.24%)	<0.001
Fraction	Yes	365 (4.71%)	310 (4.52%)	55 (6.15%)	0.036
Urea (mg/dL)		32 (26, 38)	31 (26, 37)	37 (31, 44)	<0.001
Ua (mg/dL)		4.96 (3.98, 6.05)	4.84 (3.90, 5.92)	5.85 (4.97, 6.96)	<0.001
CRP (mg/L)		1.00 (0.00, 2.00)	1.00 (0.00, 2.00)	2.00 (1.00, 4.00)	<0.001
eGFR (mL/min/1.73 m^2^)		76 (67, 85)	78 (70, 87)	55 (49, 58)	<0.001
PRAL (mEq/d)		21 (13, 28)	21 (13, 28)	21 (13, 28)	0.808
NEAP (mEq/d)		78 (65, 92)	77 (65, 92)	80 (67, 95)	<0.001

Notes: PRAL: potential renal acid load; NEAP: net endogenous acid production.

**Table 2 nutrients-16-02461-t002:** Energy and nutrient intakes (energy-adjusted, per 1000 kcal).

Nutrients	Overall (*n* = 7749)	Non-CKD (*n* = 6854)	CKD (*n* = 895)	*p*
Energy (Kcal)	1656 (1354, 1985)	1659 (1355, 1988)	1634 (1344, 1967)	0.238
Carbohydrate (g)	152 (129, 176)	153 (129, 177)	150 (127, 172)	0.006
Protein (g)	37 (32, 44)	37 (32, 44)	37 (33, 45)	0.736
Fat (g)	25 (15, 35)	25 (15, 35)	26 (17, 36)	0.004
Cholesterol (mg)	209 (101, 343)	210 (102, 343)	202 (95, 342)	0.774
Carbohydrate (g)	152 (129, 176)	153 (129, 177)	150 (127, 172)	0.006
Calcium (mg)	180 (142, 241)	179 (141, 240)	185 (147, 249)	0.006
Phosphorus (mg)	536 (478, 602)	536 (480, 602)	524 (463, 599)	0.002
Potassium (mg)	922 (777, 1107)	924 (780, 1109)	906 (744, 1103)	0.012
Magnesium (mg)	154 (133, 182)	155 (134, 183)	150 (128, 177)	0.001
Water (g)	489 (385, 620)	490 (386, 620)	484 (381, 618)	0.349

Note: variables are expressed as median (IQR).

**Table 3 nutrients-16-02461-t003:** ORs and 95% CI for CKD risk based on PRAL.

PRAL	Model 1		Model 2		Model 3		Model 4		Model 5	
	OR (95% CI)	*p*	OR (95% CI)	*p*	OR (95% CI)	*p*	OR (95% CI)	*p*	OR (95% CI)	*p*
Q1	1 (ref)		1 (ref)		1 (ref)		1 (ref)		1 (ref)	
Q2	1.02 (0.84, 1.25)	0.839	1.04 (0.82, 1.32)	0.723	1.03 (0.84, 1.26)	0.778	1.04 (0.82, 1.32)	0.715	1.02 (0.79, 1.31)	0.887
Q3	1.08 (0.89, 1.31)	0.452	1.15 (0.91, 1.45)	0.244	1.08 (0.88, 1.32)	0.462	1.15 (0.91, 1.45)	0.243	1.11 (0.85, 1.45)	0.441
Q4	1.02 (0.84, 1.24)	0.844	1.15 (0.91, 1.46)	0.247	0.98 (0.80, 1.21)	0.873	1.14 (0.90, 1.45)	0.275	1.08 (0.77, 1.52)	0.665

Notes: model 1: without covariate adjustment; model 2: adjusted for demographic data; model 3: adjusted for hypertension, diabetes, heart disease, and fraction; model 4: adjusted for hypertension, diabetes, heart disease, and fraction based on model 2; model 5: adjusted for energy, fat, cholesterol, CHO, and water.

**Table 4 nutrients-16-02461-t004:** ORs and 95% CI for CKD risk based on NEAP.

NEAP	Model 1		Model 2		Model 3		Model 4		Model 5	
	OR (95% CI)	*p*	OR (95% CI)	*p*	OR (95% CI)	*p*	OR (95% CI)	*p*	OR (95% CI)	*p*
Q1	1 (ref)		1 (ref)		1 (ref)		1 (ref)		1 (ref)	
Q2	1.04 (0.85, 1.28)	0.713	1.07 (0.84, 1.36)	0.584	1.07 (0.87, 1.32)	0.531	1.09 (0.86, 1.39)	0.482	1.07 (0.83, 1.37)	0.6
Q3	1.18 (0.97, 1.45)	0.101	1.33 (1.05, 1.69)	0.017	1.19 (0.97, 1.47)	0.097	1.32 (1.04, 1.67)	0.023	1.33 (1.03, 1.72)	0.029
Q4	1.37 (1.13, 1.67)	0.002	1.32 (1.04, 1.66)	0.02	1.34 (1.09, 1.64)	0.005	1.30 (1.03, 1.64)	0.03	1.33 (1.01, 1.76)	0.041

Notes: model 1: without covariate adjustment; model 2: adjusted for demographic data; model 3: adjusted for hypertension, diabetes, heart disease, and fraction; model 4: adjusted for hypertension, diabetes, heart disease, and fraction based on model 2; model 5: adjusted for energy, fat, cholesterol, CHO, and water.

## Data Availability

This study involved the examination of datasets that are accessible to the public. The relevant data can be located at the following source: https://www.cpc.unc.edu/projects/china/data (accessed on 1 August 2023).

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
