# Peer review of "Association between Dietary Acid Load and Chronic Kidney Disease in the Chinese Population: A Comprehensive Analysis of the China Health and Nutrition Survey (2009)"

_nutrients, 2024, doi:10.3390/nu16152461_

Round 1

Reviewer 1 Report

Comments and Suggestions for Authors

I reviewed with interest the manuscript titled "Association between Dietary Acid Load and CKD in the Chinese Population: A Comprehensive Analysis of the China Health and Nutrition Survey (2009)" which examines the relationship between dietary acid load and chronic kidney disease in a Chinese population. The study used data from the China Health and Nutrition Survey. The research question is interesting and relevant, focusing on a population-specific examination of dietary influences on CKD, an area with limited existing data for Chinese populations. The language is clear and fluid. The methods are coherent and well-detailed, with appropriate statistical analyses. The results are soundly presented, showing that while PRAL was not significantly associated with CKD, higher NEAP levels were independently linked to CKD, especially among certain subgroups. The discussion and conclusions are sufficiently balanced. 

Major Issues

The cross-sectional design limits the ability to establish causality between DAL and CKD. 

Potential recall bias in dietary intake data could affect the accuracy of DAL estimations.

Author Response

Jun 17, 2024

Dear Editors and Reviewers

On behalf of all the authors, I would like to thank the reviewers and the editors for their thoughtful comments and suggestions to our manuscript by Wang et al. (Manuscript number: 3096625) entitled “Association between Dietary Acid Load and CKD in the Chinese Population: A Comprehensive Analysis of the China Health and Nutrition Survey (2009)”. We are pleased to note that the reviewers have found our work interesting and pointed out some comments, which are all valuable and helpful for revising and improving our study, as well as the crucial guiding significance to the research.

We have thoroughly considered all the comments and tried our best to modify them following the reviewers' suggestions and correct the language duplications. We addressed the comments with point-by-point responses and highlighted the new manuscript in red.

We appreciate the time and effort you put into evaluating our work to improve our manuscript to match the readers of your high-reputation journal.

We look forward to your decision and any contact needs.

Sincerely,

Limeng Chen, MD, Ph.D

Professor of Medicine

Chief of Nephrology Division

Peking Union Medical College Hospital

Peking Union Medical College, Chinese Academy of Medical Sciences

No. 1, Shuaifuyuan, Wangfujing St, Beijing, China, 100730

Phone: +8610-69155351

Email: chenlimeng@pumch.cn

Reviewers’ Comments:

Referee #1:

General Comments:

I reviewed with interest the manuscript titled "Association between Dietary Acid Load and CKD in the Chinese Population: A Comprehensive Analysis of the China Health and Nutrition Survey (2009)" which examines the relationship between dietary acid load and chronic kidney disease in a Chinese population. The study used data from the China Health and Nutrition Survey. The research question is interesting and relevant, focusing on a population-specific examination of dietary influences on CKD, an area with limited existing data for Chinese populations. The language is clear and fluid. The methods are coherent and well-detailed, with appropriate statistical analyses. The results are soundly presented, showing that while PRAL was not significantly associated with CKD, higher NEAP levels were independently linked to CKD, especially among certain subgroups. The discussion and conclusions are sufficiently balanced.

R: Thank you so much for your kind comments.

Q1: The cross-sectional design limits the ability to establish causality between DAL and CKD.

R: Thank you for your comments. We absolutely agreed with your that the conclusion based on cohort is better than cross-sectional design and would like to focused it in our CKD cohort all over the country. We added it in the limitation.

Line 272-273:

Firstly, in this cross-sectional study, we could not prove the cause-and-effect relations between DAL and CKD.

Q2: Potential recall bias in dietary intake data could affect the accuracy of DAL estimations.

R2: Thank you for your comments and kind reminder. We agree with your concern regarding the potential recall bias in dietary intake data and its impact on the accuracy of DAL estimations. To minimize the recall bias, our colleagues explored various methods during the data collection process of the China Health and Nutrition Survey (2009). 1) We used the Food Frequency Questionnaire (FFQ) to collect information on respondents' dietary habits over a specified period to enhance its reliability. 2) We provided standardized questionnaires and coding systems with repeated measurements to reduce human error in data collection and processing. Multiple dietary recalls and surveys could capture individual dietary intake at different time points to reduce the error of a single measurement. 3) The systemic training and education of the population and the researchers contribute to more accurate remembering, recording, and measurement of food intake.

Nevertheless, we acknowledged the presence of recall bias in the limitations section of the article.

Line 274-276:

Thirdly, since dietary assessment came from recall of food intake by participants, recall bias could not be avoided, although we tried several ways to minimize it.

Reviewer 2 Report

Comments and Suggestions for Authors

Shurui Wang and his colleagues investigated the relationship between dietary acid intake and CKD. This study focuses on dietary acid and alkali, which have been attracting attention in recent years, and further investigates the relationship with CKD, making it a timely study. The number of participants was also large, and there were sufficient study subjects to withstand analysis.

1. It is difficult to evaluate the amount of acid in food, and it is difficult to say whether this study was able to evaluate the amount accurately enough. I would like to present some ideas or thoughts on how to accurately evaluate acid and alkali in food.

2. Foods with high alkaline ingredients are usually expensive and do not last long, so family support is also required. Did this study take into account social background such as annual income, educational background, and family? If it is difficult, please state so in the limitations.

Comments on the Quality of English Language

n/a

Author Response

Jun 17, 2024

Dear Editors and Reviewers

On behalf of all the authors, I would like to thank the reviewers and the editors for their thoughtful comments and suggestions to our manuscript by Wang et al. (Manuscript number: 3096625) entitled “Association between Dietary Acid Load and CKD in the Chinese Population: A Comprehensive Analysis of the China Health and Nutrition Survey (2009)”. We are pleased to note that the reviewers have found our work interesting and pointed out some comments, which are all valuable and helpful for revising and improving our study, as well as the crucial guiding significance to the research.

We have thoroughly considered all the comments and tried our best to modify them following the reviewers' suggestions and correct the language duplications. We addressed the comments with point-by-point responses and highlighted the new manuscript in red.

We appreciate the time and effort you put into evaluating our work to improve our manuscript to match the readers of your high-reputation journal.

We look forward to your decision and any contact needs.

Sincerely,

Limeng Chen, MD, Ph.D

Professor of Medicine

Chief of Nephrology Division

Peking Union Medical College Hospital

Peking Union Medical College, Chinese Academy of Medical Sciences

No. 1, Shuaifuyuan, Wangfujing St, Beijing, China, 100730

Phone: +8610-69155351

Email: chenlimeng@pumch.cn

Reviewer #2:

General Comments:

Shurui Wang and his colleagues investigated the relationship between dietary acid intake and CKD. This study focuses on dietary acid and alkali, which have been attracting attention in recent years, and further investigates the relationship with CKD, making it a timely study. The number of participants was also large, and there were sufficient study subjects to withstand analysis.

R: Thank you for your kind comments.

Q1: It is difficult to evaluate the amount of acid in food, and it is difficult to say whether this study was able to evaluate the amount accurately enough. I would like to present some ideas or thoughts on how to accurately evaluate acid and alkali in food.

R1: Thank you for your constructive suggestions. Although it is difficult to evaluate the amount of acid in food, some studies derived methods to estimate the DAL from measures of dietary intake[1-4]. Figure 1 shows the commonly used methods in calculating the NEAP and PRAL[4, 5].

  1. Endogenous Acid Production (EAP): This method evaluates the acid-base nature of foods by estimating the acid load produced during human metabolism.
  2. Gastrointestinal (GI) Alkali Absorption: Assessing the ability of foods to absorb alkaline substances during the digestive process.
  3. Net Endogenous Acid Production (NEAP): Calculating the impact of foods on internal acid-base balance by subtracting alkaline load from endogenous acid load.
  4. Potential Renal Acid Load (PRAL): Evaluating the effect of foods on renal acid load by calculating the difference between cations and anions present in foods.
  5. Net Acid Excretion (NAE): Assessing the contribution of foods to overall acid-base balance by calculating the difference between acidic substances excreted by the body and alkaline substances.

Figure1. Methods to estimate the dietary acid load.

Among these methods, the PRAL and NEAP are the most widely used in different dietary patterns (e.g., vegetarian, fish, and high-protein diets) on acid-base balance, with advantages over other methods, in clinical research and practice. Firstly, it is a quantitative assessment, accurately measuring the degree to which foods affect internal acid-base balance rather than relying on subjective judgments based on taste or chemical composition. Secondly, these methods are grounded in various ions in foods and estimates of the endogenous acid load produced during human metabolism. PRAL and NEAP values help individuals make informed dietary choices that align with their acid-base balance needs, particularly crucial for chronic conditions like osteoporosis and chronic kidney disease with varied acid-base balance disorders. So, PRAL and NEAP values, as internationally recognized standards, provide a relatively reliable method for evaluating the impact of food on internal acid-base balance. We added the explain in the discussion section as below.

Line 221-229:

Dietary intake is a crucial factor influencing the acid-base balance in individuals with chronic diseases. A substantial body of research has developed various methods to estimate dietary acid load based on dietary intake measurements including Endogenous acid production, Gastrointestinal (GI) alkali absorption, NEAP, Net acid excretion NAE, and PRAL[1-4]. The most prevalent methodologies for assessing acid-base balance are NEAP and the diet-dependent component of net acid excretion, referred to as PRAL[4, 5], both of which assess acid-base status based on food components such as calcium, phosphorus, magnesium, protein, and potassium. NEAP, calculated from the ratio of dietary protein to potassium content, indicates the equilibrium between acid and base precursors in the body.

Q2: Foods with high alkaline ingredients are usually expensive and do not last long, so family support is also required. Did this study take into account social background such as annual income, educational background, and family? If it is difficult, please state so in the limitations.

R2: Thank you so much for your thoughtful advice. Following your suggestion, we analyzed the participants' socioeconomic background factors including age, gender, educational background, family structure, and residential area. The results show that the relationship between NEAP and CKD remains significant regardless of whether these variables are adjusted for.

Upon reviewing the original data, we found that the CHNS-2009 dataset lacks the variable of annual income, so it was not included in the analysis. In addition, high-alkaline foods (such as vegetables and fruits) are not expensive and are readily available in China. Therefore, we believe that annual income may have a limited impact on dietary acid load. In future research, we will pay more attention to the impact of economic status on dietary acid load and CKD.

We will explicitly state this in the limitations section of the study. Thank you again for your valuable comments.

Line: 276-279

Fourthly, due to the presence of some missing data (sampling framework, weight information, serum calcium, serum phosphorus) in this population-based studies, the conclusions may have certain limitations when applied to a broader population, particularly the Chinese population.

Reference:

  1. Scialla JJ, Anderson CA: Dietary acid load: a novel nutritional target in chronic kidney disease? (1548-5609 (Electronic)).
  2. Sebastian A, Frassetto La Fau - Sellmeyer DE, Sellmeyer De Fau - Merriam RL, Merriam Rl Fau - Morris RC, Jr., Morris RC, Jr.: Estimation of the net acid load of the diet of ancestral preagricultural Homo sapiens and their hominid ancestors. (0002-9165 (Print)).
  3. Frassetto LA, Lanham-New Sa Fau - Macdonald HM, Macdonald Hm Fau - Remer T, Remer T Fau - Sebastian A, Sebastian A Fau - Tucker KL, Tucker Kl Fau - Tylavsky FA, Tylavsky FA: Standardizing terminology for estimating the diet-dependent net acid load to the metabolic system. (0022-3166 (Print)).
  4. Remer T, Manz F: Estimation of the renal net acid excretion by adults consuming diets containing variable amounts of protein. (0002-9165 (Print)).
  5. Remer T, Manz F: Potential renal acid load of foods and its influence on urine pH. (0002-8223 (Print)).
  6. Hu L, Napoletano A, Provenzano MA-OX, Garofalo C, Bini C, Comai G, La Manna G: Mineral Bone Disorders in Kidney Disease Patients: The Ever-Current Topic. LID - 10.3390/ijms232012223 [doi] LID - 12223. (1422-0067 (Electronic)).
  7. Lishmanov A, Dorairajan S Fau - Pak Y, Pak Y Fau - Chaudhary K, Chaudhary K Fau - Chockalingam A, Chockalingam A: Elevated serum parathyroid hormone is a cardiovascular risk factor in moderate chronic kidney disease. (1573-2584 (Electronic)).

Reviewer 3 Report

Comments and Suggestions for Authors

  This is a very well written paper the hypothesis is clear and the findings of the study are an important contribution to the literature.  Additionally the findings suggest that dietary interventions may impact the development of CKD in Asian populations

The study investigators employing This study employed data from the China Health and Nutrition Survey.

evaluating the effects of a daily dietary acid load on the risk of chronic kidney disease evaluating the association of potential acid load and net endogenous acid excretion and CKD.  Employing univariate and multivariate analysis the study demonstrated that as net endogenous acid production increases so does the risk of CKD.  This association remains significant after adjusting for several factors associated with CKD.

This is an important study and contribute to the current evidence of risk factors for CKD particularly in Aian populations.

Minor revisions

 This study employed data from the China Health and Nutrition Survey.  For example where what areas/regions in China were the population surveyed?  The investigators need to give more details on how this survey is conducted or reference where further details can be obtained.

Of importance if the investigators findings can be applied to the Chinese population shouldn’t the evaluation of the  data obtained from The China Health and Nutrition Survey be a weighted analysis. Additionally this data should be evaluated using methods that adjust for survey analysis and employs complex samples analysis. In my opinion this would greatly impact the importance of their findings.  

Some additional studies should be considered such as adjusting for calcium phosphate product

Can the investigators comment on the effects of sarcopenia and net endogenous acid production was most profound in the elderly and those with smaller waist circumference?

Author Response

Jun 17, 2024

Dear Editors and Reviewers

On behalf of all the authors, I would like to thank the reviewers and the editors for their thoughtful comments and suggestions to our manuscript by Wang et al. (Manuscript number: 3096625) entitled “Association between Dietary Acid Load and CKD in the Chinese Population: A Comprehensive Analysis of the China Health and Nutrition Survey (2009)”. We are pleased to note that the reviewers have found our work interesting and pointed out some comments, which are all valuable and helpful for revising and improving our study, as well as the crucial guiding significance to the research.

We have thoroughly considered all the comments and tried our best to modify them following the reviewers' suggestions and correct the language duplications. We addressed the comments with point-by-point responses and highlighted the new manuscript in red.

We appreciate the time and effort you put into evaluating our work to improve our manuscript to match the readers of your high-reputation journal.

We look forward to your decision and any contact needs.

Sincerely,

Limeng Chen, MD, Ph.D

Professor of Medicine

Chief of Nephrology Division

Peking Union Medical College Hospital

Peking Union Medical College, Chinese Academy of Medical Sciences

No. 1, Shuaifuyuan, Wangfujing St, Beijing, China, 100730

Phone: +8610-69155351

Email: chenlimeng@pumch.cn

Reviewer #3:

General Comments:

This is a very well written paper the hypothesis is clear and the findings of the study are an important contribution to the literature. Additionally, the findings suggest that dietary interventions may impact the development of CKD in Asian populations. The study investigators employing This study employed data from the China Health and Nutrition Survey evaluating the effects of a daily dietary acid load on the risk of chronic kidney disease evaluating the association of potential acid load and net endogenous acid excretion and CKD. Employing univariate and multivariate analysis the study demonstrated that as net endogenous acid production increases so does the risk of CKD. This association remains significant after adjusting for several factors associated with CKD. This is an important study and contribute to the current evidence of risk factors for CKD particularly in Aian populations.

R: Thank you for your kind comments.

Q1: This study employed data from the China Health and Nutrition Survey.  For example, where what areas/regions in China were the population surveyed?  The investigators need to give more details on how this survey is conducted or reference where further details can be obtained.

Response 1: Thank you for your reminder and suggestions. We added the background and detailed information of the China Health and Nutrition Survey in the method as below.

Line 76-79:

This study utilized openly accessible data from the CHNS. This study utilized a multi-stage random clustering method to gather comprehensive data on the socioeconomic status, healthcare utilization, dietary habits, and nutritional status of a consistent cohort over time. Finally, we selected approximately 7,200 households, with a total population exceeding 30,000 individuals.

Question 2: Of importance if the investigators findings can be applied to the Chinese population shouldn’t the evaluation of the data obtained from The China Health and Nutrition Survey be a weighted analysis. Additionally, this data should be evaluated using methods that adjust for survey analysis and employs complex samples analysis. In my opinion this would greatly impact the importance of their findings. 

Response 2: We appreciate your concern. We fully understand and acknowledge the importance of weighted analyses and complex sample analysis methods. These methods can correct biases caused by sampling design, thereby improving the accuracy of data analysis and the reliability of results. However, due to restrictions in data sources, we cannot obtain a complete sampling framework and weight information, making it difficult to perform weighted and complex sample analyses.

Despite these constraints, we have conducted thorough analyses within the current conditions to ensure that our research findings are as accurate and reliable as possible given the existing data conditions.

We sincerely acknowledge that due to the above data limitations, our research conclusions may have certain limitations when applied to a broader population, particularly the Chinese population. We have clearly pointed out these limitations in our discussion section. Additionally, we suggest that future research should obtain more comprehensive and high-quality data and adopt weighted analyses and complex sample analysis methods to further validate and expand our findings.

We greatly appreciate your valuable feedback, which will help improve and enhance our future research endeavors. We hope our response meets your requirements, and please feel free to contact us if you have any further questions or suggestions.

Line 276-279:

Fourthly, due to the presence of some missing data (sampling framework and weight information) in this population-based studies, the conclusions may have certain limitations when applied to a broader population, particularly the Chinese population.

Q3: Some additional studies should be considered such as adjusting for calcium phosphate product.

R3: We appreciate your concern.

R3: Thank you for your suggestion. We agreed that the calcium-phosphorus product (Ca × P) could provide important information indicating an imbalance in the patient’s mineral metabolism and predict the prognosis of CKD patients with increased mortality and cardiovascular events in CKD patients  [6] [7]. Unfortunately, the CHNS did not collect the blood phosphorus and calcium levels during the sample collection process in 1989, when the crucial of CKD-MBD was not recognized. We added it to the limitation.

Line 276-279:

Fourthly, due to the presence of some missing data (sampling framework, weight information, serum calcium, serum phosphorus) in this population-based studies, the conclusions may have certain limitations when applied to a broader population, particularly the Chinese population.

Question 4: Can the investigators comment on the effects of sarcopenia and net endogenous acid production was most profound in the elderly and those with smaller waist circumference?

R4: Thank you for your insightful question. We conducted sensitivity analyses that indicated that the positive correlation between NEAP and CKD appeared stronger in those aged ≥60 years, with lower waist circumference and BMI< 24 kg/m2. We added this finding in the result and discussion below.

Result Section

The sensitivity analysis results suggested that the positive correlation between NEAP and CKD appeared stronger in those aged ≥60 years, with lower waist circumference and BMI< 24 kg/m2.

Discussion Section

Line 245-254

In this study, we found that older adults and individuals with smaller waist circumferences tended to have higher acid load. Because sarcopenia, the loss of muscle mass and function, can increase the body's acid load by disrupting the neutralization of acid produced during the metabolism of amino acids in muscle tissue. Additionally, people with smaller waist circumferences often have less muscle mass and fat reserves, which can lead to disturbances in acid-base balance and are more susceptible to the effects of NEAP. These findings underscore the significance of body composition and muscle mass in evaluating dietary acid load and its health repercussions, particularly in vulnerable elderly groups.

Round 2

Reviewer 2 Report

Comments and Suggestions for Authors

The revised manuscript is well written.  I think this manuscript is worth publishing.